# Predicting Ovarian-Cancer Burden in Catalonia by 2030: An Age–Period–Cohort Modelling

**DOI:** 10.3390/ijerph19031404

**Published:** 2022-01-27

**Authors:** Paula Peremiquel-Trillas, Jon Frias-Gomez, Laia Alemany, Alberto Ameijide, Mireia Vilardell, Rafael Marcos-Gragera, Sònia Paytubi, Jordi Ponce, José Manuel Martínez, Marta Pineda, Joan Brunet, Xavier Matías-Guiu, Marià Carulla, Jaume Galceran, Ángel Izquierdo, Josep M. Borràs, Laura Costas, Ramon Clèries

**Affiliations:** 1Cancer Epidemiology Research Programme, Catalan Institute of Oncology, Av. Gran Vía 199-203, L’Hospitalet de Llobregat, 08908 Barcelona, Spain; paula.peremiquel@iconcologia.net (P.P.-T.); jfrias_ext@iconcologia.net (J.F.-G.); lalemany@iconcologia.net (L.A.); spaytubic@iconcologia.net (S.P.); 2Bellvitge Biomedical Research Institute-IDIBELL, Av. Gran Vía 199-203, L’Hospitalet de Llobregat, 08908 Barcelona, Spain; jponce@bellvitgehospital.cat (J.P.); jmartinezgar@bellvitgehospital.cat (J.M.M.); Jbrunet@iconcologia.net (J.B.); xmatias@bellvitgehospital.cat (X.M.-G.); jmborras@iconcologia.net (J.M.B.); 3Faculty of Medicine, University of Barcelona, C/Casanova 143, 08036 Barcelona, Spain; 4Consortium for Biomedical Research in Epidemiology and Public Health-CIBERESP, Carlos III Institute of Health, Av. De Monforte de Lemos 5, 28029 Madrid, Spain; rmarcos@iconcologia.net; 5Tarragona Cancer Registry, Cancer Epidemiology and Prevention Service, Sant Joan de Reus University Hospital, Av. Dr. Josep Laporte 2, 43204 Reus, Spain; alberto.ameijide@salutsantjoan.cat (A.A.); maria.carulla@salutsantjoan.cat (M.C.); jaume.galceran@salutsantjoan.cat (J.G.); 6Pere Virgili Health Research Institute-IISPV, Rovira i Virgili University, Av. de la Universitat, 1, 2ª pl. Reus, 43204 Tarragona, Spain; 7Independent Researcher, 08028 Barcelona, Spain; mvilardelln@uoc.edu; 8Epidemiology Unit and Girona Cancer Registry, Oncology Coordination Plan, Catalan Institute of Oncology, Department of Health, Government of Catalonia, Av. França-Sant Ponç s/n, 17007 Girona, Spain; aizquierdo@iconcologia.net; 9Descriptive Epidemiology, Genetics and Cancer Prevention Research Group, Girona Biomedical Research Institute-IDIBGI. C/Dr. Castany s/n. Edifici M2, Parc Hospitalari Martí i Julià, 17190 Salt, Spain; 10Medical Sciences Department, Faculty of Medicine, University of Girona, C/Emili Grahit 77, 17071 Girona, Spain; 11Department of Gynecology and Obstetrics, Bellvitge University Hospital, Feixa Llarga s/n, L’Hospitalet de Llobregat, 08907 Barcelona, Spain; 12Hereditary Cancer Programme, Catalan Institute of Oncology, Av. Gran Vía 199-203, L’Hospitalet de Llobregat, 08908 Barcelona, Spain; mpineda@iconcologia.net; 13Consortium for Biomedical Research in Cancer-CIBERONC, Carlos III Institute of Health, Av. De Monforte de Lemos 5, 28029 Madrid, Spain; 14Medical Oncology Department, Catalan Institute of Oncology, Doctor Josep Trueta Girona University Hospital, Av. França-Sant Ponç s/n, 17007 Girona, Spain; 15Department of Pathology, Bellvitge University Hospital, Feixa Llarga s/n, L’Hospitalet de Llobregat, 08907 Barcelona, Spain; 16Statistical Section, Genetics, Microbiology and Statistics Department, Faculty of Biology, University of Barcelona, Av. Diagonal 643, 08028 Barcelona, Spain; 17Oncology Coordination Plan, Catalan Institute of Oncology, Department of Health, Government of Catalonia, Av. Gran Vía 199-203, L’Hospitalet de Llobregat, 08908 Barcelona, Spain; 18Clinical Sciences Department, Faculty of Medicine, University of Barcelona, Feixa Llarga s/n, L’Hospitalet de Llobregat, 08907 Barcelona, Spain

**Keywords:** ovarian cancer, projections, burden, incidence, mortality, survival, time trends

## Abstract

Ovarian cancer is the most lethal gynaecological cancer in very-high-human-development-index regions. Ovarian cancer incidence and mortality rates are estimated to globally rise by 2035, although incidence and mortality rates depend on the region and prevalence of the associated risk factors. The aim of this study is to assess changes in incidence and mortality of ovarian cancer in Catalonia by 2030. Bayesian autoregressive age–period–cohort models were used to predict the burden of OC incidence and mortality rates for the 2015–2030 period. Incidence and mortality rates of ovarian cancer are expected to decline in Catalonia by 2030 in women ≥ 45 years of age. A decrease in ovarian-cancer risk was observed with increasing year of birth, with a rebound in women born in the 1980s. A decrease in mortality was observed for the period of diagnosis and period of death. Nevertheless, ovarian-cancer mortality remains higher among older women compared to other age groups. Our study summarizes the most plausible scenario for ovarian-cancer changes in terms of incidence and mortality in Catalonia by 2030, which may be of interest from a public health perspective for policy implementation.

## 1. Introduction

Ovarian cancer (OC) is the ninth most common cancer type among women worldwide in terms of incidence and the eighth in terms of mortality [1]. In regions with a very high human-development index, OC is the fifth most common cancer in terms of number of deaths among women and the most lethal gynaecological cancer [1]. OC has the lowest survival rate of all gynaecological cancers; worldwide, the estimated survival of five years is 46% [2]. Due to the non-specificity of its symptoms, most of the cases are diagnosed at an advanced stage [3,4]. OC classification includes different histologic types that present notable differences in terms of incidence, origin, pathogenesis, gene expression, molecular alterations and prognosis [5]. Epithelial tumours are the most frequent type of OC, as they account for over 90% of the cases, while sex-cord stromal and germ-cell tumours represent 5–6% and 2–3% of cases, respectively [2].

The main risk factor for OC is age, and over 50% of the cases occur in women over 65 years old [6]. Genetic predisposition is responsible for up to 15% of the cases; in particular, BRCA-mutation carriers (65–75%) and Lynch-syndrome (10–15%) patients have a higher risk of OC [2]. A decrease in the number of lifetime ovulations and the use of postmenopausal hormonal therapy (HT) have also been significantly associated with an increased OC risk [2,7]. On the other hand, breastfeeding and the use of hormonal contraceptives have been associated with OC-risk reduction [7,8]. Benign gynaecological conditions such as endometriosis or pelvic inflammatory disease have been associated with an increased risk of specific OC histologic subtypes, while some gynaecological procedures (tubal ligation, hysterectomy, and prophylactic oophorectomy) have been shown to be associated with a reduced risk [2,7,9]. Certain lifestyle factors such as tobacco use or increased body-mass index (BMI) have also been associated with a higher risk of specific OC histologic subtypes [2,7,9]. Globally, the OC-incidence rate and the associated number of deaths are estimated to rise by 2035, mainly because of an increasing burden of disease in low- and middle-income countries [10]. Nevertheless, a reduction in OC incidence and mortality rates has been observed in some regions over recent years, such as in the USA and regions of Europe [11,12]. Evidence suggests that this reduction has mostly been related to the increased use of hormonal contraceptives and the extension of genetic counselling with an optimized follow-up of those women at a higher OC risk, such as BRCA-mutation carriers and women with Lynch Syndrome, as well as health-system improvements [13,14].

In Catalonia, the OC incidence is similar to that of other developed countries: it was the eighth most common cancer among the female population in 2015 and it was responsible for 2.55% of the incidence of cancer cases among women in 2018, with a 5-year-survival rate between 41.9% and 49.3% among those women diagnosed with OC during the period of 2010–2014 [15]. A decrease in OC burden has been observed in this region in the last decade [15]. This work aims to inquire more about this trend in order to predict changes in OC incidence and mortality in Catalonia by 2030, and to discuss their biological plausibility. Given that mortality depends on incidence and survival, we also evaluated the time trends of ovarian-cancer patients’ survival in order to assess a potential change in mortality trends.

## 2. Materials and Methods

### 2.1. Data

Ovarian-cancer incidence in Catalonia was estimated using data from the two population-based cancer registries in Catalonia for the period between 1994 and 2012. Population-based cancer registries monitor cancer incidence in a geographically defined area by recording all cancer-incidence cases. In Spain, 13 cancer registries currently monitor all cancer sites, covering 26% of the Spanish population [15,16]. The two population-based cancer registries in Catalonia cover 20% of the population [15]. According to the previous literature, the ICD-10-CM diagnosis code used was C56, which refers to a malignant neoplasm of the ovary [17]. Cancer-mortality data for the period 1994–2013 were obtained from the Catalan mortality registry, which covers the whole Catalan population. Once fitted to the model, the incidence and mortality data were aggregated in annual intervals in 5-year age groups, establishing a cut-off at 45 years of age. Data on the Catalan population by 5-year age groups were provided by the Catalan Institute of Statistics (http://www.idescat.cat, accessed on 14 December 2021). Data for the period spanning from 1994 to 2017 was observed, while data for the 2018–2030 period was projected.

### 2.2. Statistical Modelling

The cancer incidence in Catalonia from 1997 to 2012 was estimated by applying age-specific cancer rates from the population-based cancer registries in Catalonia (Girona and Tarragona). Based on these estimates, Bayesian autoregressive age–period–cohort models were fitted to the data from the period between 1997 and 2012. These models were used to predict the burden of OC incidence and mortality rates for the 2015–2030 period for the entire cohort of women. Cancer incidence modelling was performed in two steps. First, we applied the age distribution of the Catalan population during 1997–2012 to the age-specific incidence rates of Girona and Tarragona, and then we obtained the number of ovarian cases in Catalonia for this period. Second, we fitted an age–period–cohort model to this data and used it to project the cancer incidence in Catalonia up to 2030. However, for the mortality data, mortality rates were available for the whole territory; therefore, projections of mortality up to 2030 were directly derived from the model. We also assessed the percentage change in incident cases between 2015 and 2025 by using RiskDiff [18]. RiskDiff implements the Bashir–Esteve method [19] to split the net change (NC) in the number of cases into changes in demography and changes in cancer risk (R). Since demographic changes can be divided into changes in population size (S) and population structure (Aging: A), the NC can be partitioned into 3 additive quantities: NC = A + R|A + S|A, where changes in risk and population size are conditioned by changes in population structure (R|A and S|A) [19]. Details of this statistical modelling are described elsewhere [15]. The cut-off age for OC-burden estimates in 2015 and the projections for 2030 were established at 45 years since the incidence of OC in women under 45 years old is very low. Age-standardized rates (ASRs) were reported to the European Standard Population [20]. Age-specific rates of OC incidence and mortality rates in 2012 and 2030 were also compared. A 95% prediction interval was provided for rates beyond 2012.

## 3. Results

Time trends of OC incidence and mortality by birth cohort and period of diagnoses during 1997–2027 are described in Figure 1. A decrease in OC risk was observed with increasing birth year, with a rebound in women born in the 1980s (Figure 1a). Similarly, a decrease in the risk of death was observed among those born after the 1960s, with stabilization afterwards (Figure 1b). However, the impact of the period of diagnosis in the time trends for OC incidence and death remained stable with a levelling off in the period between 1997 and 2018, which is predicted to continue until 2027 (Figure 1c,d).

Age-standardized OC-incidence and mortality rates (per 100,000 women-years, standardized to the European population) among women aged ≥ 45 years showed gradual reductions until 2012 (Annual average percentage change (AAPC): Incidence = −1.7%; Mortality = −1.1%), and these decreasing trends are estimated to be more pronounced during 2013–2030 (AAPC: Incidence = −2.7%; Mortality = −2.1%) (Figure 2a). Projections of age-specific incidence and mortality rates of OC in 2030 with respect to the reference year of 2012 show a decline in OC rates, both in terms of incidence and mortality (Figure 2b), which are expected in all age groups, in particular among older women.

Table 1 shows the OC-burden estimates for the cases and deaths in 2015 and the projections for 2030 according to age groups. Overall, a reduction in the incident OC cases and deaths (21.80% and 10.53%, respectively) is expected by 2030 among all age groups compared to the 2015 estimates. OC mortality is substantially higher in women aged ≥ 75 years than other groups and deaths exceed the incident cases in this age group. Worth noting is that the −21.8% net change in the number of cases of incidence between 2015 and 2030 can be mainly attributed to the decrease in the risk of developing OC, since −34.6% is due to changes in the risk of developing OC, 12% due to changes in population structure (aging) and 0.8% due to changes in population size (NC = R + A + S; −21.8% = −34.6% + 12% + 0.8%).

## 4. Discussion

Our study shows a gradual decrease in OC incidence and mortality rates among all age groups in Catalonia by 2030. Decreases in incidence and mortality rates are observed consistently by birth cohort as well as by period of diagnosis and period of death. However, women born in the 1980s show a higher risk of developing OC than adjacent cohorts. Despite the decline, OC mortality remains higher in 2030 among older women compared to other age groups.

The observed decline in OC risk and mortality over time has also been described in other developed countries [12,21,22,23,24,25]. The reduction in OC risk could be attributed to the changing trends in hormonal factors, which are well-established risk factors for OC. In particular, a decline in HT use and the increase in the use of hormonal contraceptives have been observed in recent decades in developed countries [23,24,26]. The use of HT decreased worldwide after 2002, subsequently to the publication of the Women’s Health Initiative report [26], and this decline has also been described among Spanish women [27]. Moreover, a growing use of hormonal contraceptives has been reported in recent decades. Nearly 50% of Spanish women stated using hormonal contraception at some point in their lives, and younger cohorts reported more extensive periods of use compared to women born before 1950 [27]. The OC-mortality reduction is probably aligned with the advances in diagnosis, treatment and management of OC in recent decades [21].

Nevertheless, the upswing in OC risk among women born in the 1980s is not likely to be explained by the factors mentioned above. Spanish women have experienced a decrease in fertility which has contributed, along with other factors, to a lower parity and increased use of fertility treatments, which seem to be associated with a higher OC risk [28]. Obesity is dramatically rising worldwide [29], and an association between obesity and OC has been observed in some of the less common OC subtypes, such as low-grade/borderline serous OC [30]. Changes in the incidence of certain OC subtypes could contribute to a change in their relative distribution, and the expected increase in this cohort of women could be related to an increase in a specific subtype [12,24]. A better understanding of OC-subtype incidences will allow us to assess specific changes in relation to risk factors. Hence, it would be of great interest for cancer registries to record cancer subtypes and not only cancer types. An increased BMI and lower parity are established risk factors for OC and influence OC incidence [31,32]. However, since BMI and fertility rates did not substantially change in Spain in recent years [32,33,34], and because this increase is not observed in the successive cohorts, other factors should be considered. Genetic counselling, for example, has contributed to the decline in the hereditary forms of OC by performing prophylactic surgeries on BRCA-mutation carriers and Lynch-syndrome patients [14] as well as improvements in gynaecological care and health systems in general.

The larger percentage of OC mortality in older women (≥75 years) compared to other age groups could be attributed to an extended survival time due to the improvement of OC treatment and management in recent years among women diagnosed with OC at younger ages [16]. It could also be attributed to the detection of OC at earlier stages due to faster access to the health system and increased attendance for gynaecological check-ups, which could result in a more prolonged survival. Notwithstanding, the predicted stabilization in mortality rates in the near future and the low survival of this cancer highlights the importance of promoting research towards screening and early detection of OC. Although enormous advances in OC screening and early detection have been made in recent decades, those advances have not been demonstrated to improve OC survival [35,36,37].

The strength of our study is its use of Bayesian methodology for the construction of the predictive models, providing robust estimates for OC incidence and mortality in Catalonia by 2030. However, some limitations should be considered when interpreting the results. Mortality data were available from the entire Catalan territory, while incidence data were retrieved from two existing regional population-based cancer registries in Catalonia. In Spain, mortality data are collected across all provinces, whereas cancer-incidence data are not, since cancer registration in Spain is not mandatory in all regions and provinces [16,25]. Cancer incidence in Spain is estimated by modelling and using cancer-incidence and mortality data aggregated by province [25]. REDECAN, the network of Spanish population-based cancer registries (http://redecan.org/es/index.cfm, accessed on 20 January 2022) provides this valuable information about the estimation of the burden of cancer in Spain [25]. To that end, the estimation of ovarian-cancer incidence in Catalonia is based on the information provided by the two population-based cancer registries in this region, Girona and Tarragona, both members of REDECAN. Since no differences among these two registries were observed, we assumed homogeneous OC-incidence rates throughout the Catalan territory [15], although this assumption may misestimate OC-incident cases.

## 5. Conclusions

Our study summarizes the most plausible scenario of ovarian-cancer changes in terms of incidence and mortality rates in Catalonia by 2030 according to the available data. Ovarian-cancer incidence and mortality rates are expected to decline in Catalonia by 2030, probably due to the increasing hormonal-contraceptive use, the decline of HT and improvements in the healthcare systems. This information may be of interest from a public health perspective to health-planning decisions and may allow the implementation of policies according to this scenario.

## Figures and Tables

**Figure 1 ijerph-19-01404-f001:**
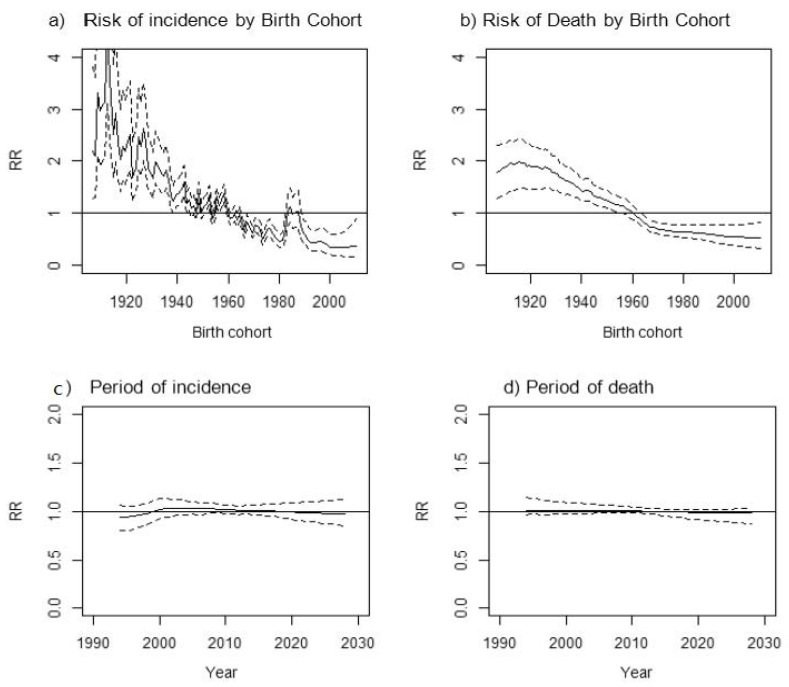
(**a**,**b**) Time trends of incidence/mortality for the entire cohort of women: risk of incidence/death by birth cohort; (**c**,**d**) period of diagnosis for ovarian cancer in Catalonia during 1997–2027. The dashed line indicates 95% prediction intervals. Note: These figures reflect changes in the overall trend of incidence and mortality for the entire cohort of women concerning the average of rates across birth cohorts or period of diagnosis/death. For instance, cohorts born before the 1960s presented a higher risk (relative risk, RR) of ovarian cancer incidence and mortality than cohorts born later. The time trend of ovarian cancer incidence and death remained stable according to the period of diagnosis.

**Figure 2 ijerph-19-01404-f002:**
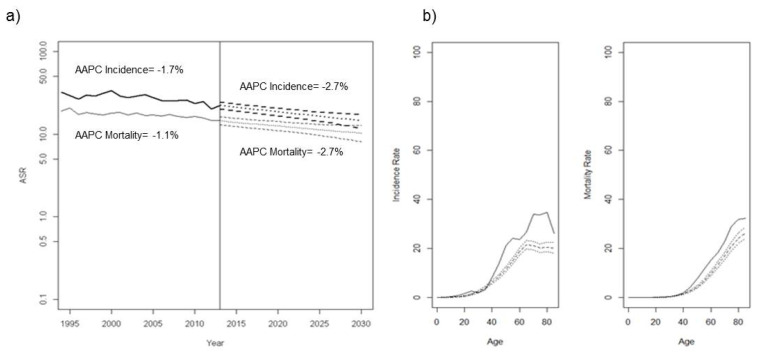
Time trends and projections of ovarian cancer rates up to 2030. (**a**) Age-standardized incidence and mortality rates (per 100,000 women-years, standardized to the European Population) of ovarian cancer among women diagnosed ≥ 45 years (thick line: incidence; thin line: mortality; dashed lines: predicted ASRs and their corresponding 95% prediction intervals); (**b**) Age-specific incidence and mortality rates of ovarian cancer in Catalonia in 2030. AAPC: Annual average percentage change (**a**) Thick line: incidence; thin line: mortality; dashed lines: predicted ASRs and their corresponding 95% prediction intervals. (**b**) solid line: rates of the reference year 2012; dashed and dotted lines: predicted rates and their corresponding 95% prediction intervals, respectively.

**Table 1 ijerph-19-01404-t001:** Ovarian-cancer-burden estimates for the number of cases and deaths in 2015 and projections for 2030 in Catalonia according to age groups, based on data from the Tarragona and Girona Cancer Registries.

	Age Groups											Net Change ^2^ 2030 vs. 2015	ASR ^3^
	<45		45–54		55–64		65–74		≥75		Total		
	N	% ^1^	N	% ^1^	N	% ^1^	N	% ^1^	N	% ^1^	N		
**Incidence 2015**	48	10.8	80	17.9	110	24.7	92	20.7	115	25.9	445		22.4
**Incidence 2030**	34	9.8	55	15.8	78	22.4	89	25.6	92	26.4	348	−21.80%	14.8
**Mortality 2015**	9	3.3	25	9.4	50	18.8	65	24.4	117	43.9	266		14.0
**Mortality 2030**	6	2.5	20	8.4	39	16.4	60	25.2	113	47.5	238	−10.53%	10.3

^1^ Percentage corresponds to row percentage. ^2^ Change: Percentage variation in the number of OC cases between 2030 and 2015. ^3^ ASR: Age—standardized rates to the European Standard Population per 100,000 women-years.

## Data Availability

The relevant data are available in the manuscript. Data that are not presented in the article are available upon reasonable request from the corresponding authors.

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
