# Peer review of "Predicting Ovarian-Cancer Burden in Catalonia by 2030: An Age–Period–Cohort Modelling"

_ijerph, 2022, doi:10.3390/ijerph19031404_

Round 1

Reviewer 1 Report

I was pleased to review this paper.

  1. The methodology used by the Authors is appropriate for the purpose of the study and conclusions are narrow linked to data discussion and available evidence. The English language is fluid and well understood. Nevertheless, the paper is of good quality, I would have added just more recent studies, for this is need a minor revision.

I suggest this:

  • Why was mortality data available from the entire Catalan territory, while was incidence data retrieved from two existing regional population-based cancer registries in Catalonia?
  • Ovarian cancer incidence and mortality is expected to decrease in Catalonia by 2030, probably due to the increase in hormonal contraceptive use, the decline of HT and improvements in health systems, in this regard, I recommend. to add a greater number of arguments than the international literature, so as to define everything in the best way.

The results are consistent with the study. Corrected the method used in the work, given the statistical results obtained.  New study scenarios are opening up for the future.

Reviewer 2 Report

The aim of this paper is order to predict changes in OC incidence and mortality in Catalonia by 2030, and describe their biological plausibility. This information may be of interest from a public health perspective, , especially in Europe.  However, the manuscript requires some corrections before being accepted. The list of comments below.

  1. Abstract:

The abstract should be without headings

107-110: Sentence ” OC has the lowes…” divide on two please

121-126: divide the sentences into gynaecological procedures or conditions, and   description of lifestyle factors

121: Replace ”con-ditions” on “conditions”

120: Replace “hormonal con-traceptives” on “ hormonal contraceptives”

130:  Replace “re-gions of” on “regions”

131: Replace “hor-monal”

Methods:

Please described more specific details of this statistical modelling . The presented article will be more readable. In your previous cited article, Cleries  R. et al. 2018,  the Bashir-Estève method , to divided the net change in the number of cases between 2015 and 2025  was used. In the  presented manuscript we have got a Bayesian autoregressive age–period–cohorts models.
